# Nurse-Administered Sedation in Digestive Endoscopy: A Systematic Review

**DOI:** 10.3390/diagnostics15081030

**Published:** 2025-04-17

**Authors:** Miriam Hidalgo-Cabanillas, José Alberto Laredo-Aguilera, Esperanza Barroso-Corroto, Ángel López-González, Joseba Rabanales-Sotos, Juan Manuel Carmona-Torres

**Affiliations:** 1Hospital Universitario de Toledo, 45004 Toledo, Spain; 2Facultad de Fisioterapia y Enfermería de Toledo, Universidad de Castilla-La Mancha, 45071 Toledo, Spain; josealberto.laredo@uclm.es (J.A.L.-A.); esperanza.barroso@uclm.es (E.B.-C.); juanmanuel.carmona@uclm.es (J.M.C.-T.); 3Grupo de Investigación Multidisciplinar en Cuidados (IMCU), Universidad de Castilla-La Mancha, 45071 Toledo, Spain; 4Instituto de Investigación Sanitaria de Castilla-La Mancha (IDISCAM), 45004 Toledo, Spain; 5Facultad de Enfermería, Universidad de Castilla-La Mancha, Campus Universitario s/n, 02071 Albacete, Spain; angel.lopez@uclm.es (Á.L.-G.); joseba.rabanales@uclm.es (J.R.-S.); 6Grupo de Actividades Preventivas en el Ámbito Universitario de Ciencias de la Salud (GAP-CS), Universidad de Castilla-La Mancha, Campus Universitario s/n, 02071 Albacete, Spain

**Keywords:** nurse anesthetists, endoscopy, nursing, sedation, endoscopy digestive system, systematic review

## Abstract

**Background/Objectives:** Nurses are becoming increasingly common healthcare professionals who perform sedation in digestive endoscopy services. Efficacy, safety and patient satisfaction are indicators of quality and safety in the administration of sedation by nurses in patients undergoing digestive endoscopy. Therefore, the aim of this study was to synthesize the scientific evidence available to date on the efficacy, safety and patient satisfaction of nurse-administered sedation during digestive endoscopies. **Methods:** A systematic review was conducted according to PRISMA standards. The PubMed, Scopus, SciELO, Web of Science and LILACS databases were consulted. Narrative synthesis and descriptive statistics were used to explore complications arising from the use of sedation by nurses, using the percentage of complications and total events in the included studies. Owing to the methodology and heterogeneity of the included studies, a meta-analysis was not possible. **Results:** A total of 292 studies were collected, 13 of which were selected. In terms of efficacy, studies indicate that nurse sedation is effective in ensuring patient comfort and optimizing procedure times. On the other hand, studies have evaluated the safety of sedation, measured complications during sedation and reported incident and minor complication rates of less than 2.5%. Finally, studies analyzing patient satisfaction and patient experience have demonstrated high levels of satisfaction. **Conclusions:** Nurse-administered sedation demonstrates high efficacy in ensuring patient comfort and optimizing procedure times. Nurse-administered propofol sedation is safe for endoscopic procedures, with low rates of significant adverse events. Patients reported high levels of satisfaction with nurse-administered sedation, with reduced discomfort and improved perceptions of the quality of the procedure.

## 1. Introduction

Gastrointestinal (GI) endoscopy is frequently used in the diagnosis and treatment of gastrointestinal disorders. This technique has proven to be an effective method for significantly reducing the mortality and incidence of gastrointestinal cancers, such as colorectal and gastric cancers [1,2]. Although these procedures are widely used to diagnose and treat digestive tract conditions, they can be painful and cause anxiety [3]. This is why sedation becomes a necessary factor by allowing greater tolerance, reducing pain and minimizing discomfort [4,5,6].

Throughout history, the administration of sedatives has been the exclusive responsibility of anesthetists, but the continuous increase in the demand for endoscopic procedures has led to an evolution in the role of nurses, who have assumed a relevant and growing role [7] and who are now actively participating in the administration of sedation under specific protocols [6,8].

As the role of nurses in the administration of sedation in digestive endoscopies has expanded, the need to provide specific training and certifications to ensure that nurses have appropriate competencies has become evident [9,10]. Another important aspect is the legal framework that surrounds this practice of sedation by nurses [11]. In fact, in some countries, trainee and trained anesthesia nurses administer sedation and monitor patients in a variety of clinical procedures [12]. This specialization can be seen in endoscopy services, where GI endoscopy procedures have become more complicated and varied, therefore requiring greater safety and quality of care [13]. In fact, endoscopy services are where nurses frequently administer sedation autonomously.

With respect to their functions and responsibilities, these nurses not only carry out an initial and continuous evaluation of the patient but also assume the management of pain with the administration of anesthesia during the procedures; they collaborate with other health professionals to guarantee the safety and health of the patient, continuously monitor vital signs, respond quickly to any complications that may arise and monitor anesthetic recovery [12,14,15,16]. In Spain, there are no anesthesia nurses, although there are clinical practice guidelines and other consensus documents from scientific societies, such as the Spanish Association of Gastroenterology (SEG) and the Spanish Society of Digestive Endoscopy (SEED), which endorse that this procedure be performed by nurses in certain cases, depending on the type of test and the level of sedation needed, who have previously received specific training for the management of complications and safety protocols [17,18,19,20].

There is variability in patients receiving sedation by nurses in digestive endoscopy services, from patients with chronic diseases such as diabetes, heart disease and chronic lung disease to acute cases. These patients may require sedation for different procedures [21,22].

In this context, international organizations, such as the WHO (World Health Organization), the International Council of Nurses, the ASA (American Society of Anesthesiologists), the ACG (American College of Gastroenterology) and the ASGE (American Society for Gastrointestinal Endoscopy), have mentioned the importance of standardizing sedation training for nurses to ensure safe and effective practices in medical centers [23,24], suggesting that such training should include detailed simulations and manuals that guide the response to adverse events and help improve safety levels in the procedure [25,26].

With respect to the quality and safety indicators of the procedure, the experience of the patient is an important element in the context of sedation administered by nurses. Previous studies involving GI endoscopies with sedation by nurses reported that empathy and the quality of the nurse–patient relationship are determining factors in patients’ perceptions of safety and comfort [6]. Various studies have shown that patients who receive sedation via digestive endoscopy services by nurses have high levels of satisfaction and low pain during the procedure [27,28,29,30,31]. The high satisfaction of patients with sedation performed by nurses reflects the efficacy and acceptance of this practice, as seen in a review of patient satisfaction with digestive endoscopies [32].

Likewise, the selection of sedative agents by nurses can influence patient satisfaction, since some sedatives provide better pain control and a shorter recovery time, key aspects in the improvement of the experience and acceptance of future procedures [31,33,34]. On the other hand, this satisfaction can also be influenced by personal and contextual factors [35], as well as by waiting times, both when obtaining the appointment and on the same day of the test [35,36,37]. This leads to a deterioration in the degree of satisfaction that can affect the overall assessment.

Another critical quality and safety indicator of the sedation procedure is the percentage of complications associated with the use of this practice by nurses. Serious complications are relatively rare but require a high level of preparation and rapid response from the nurses involved in the procedure [38]. These complications include cardiopulmonary events such as hypoventilation, respiratory depression, apnea, hypotension and bradycardia [6,15]. Continuous evaluation and evidence-based training are essential to maintain and improve nursing skills in sedation, thus ensuring the safety and well-being of patients during endoscopic procedures [39].

Efficacy, safety and patient satisfaction are critical indicators of the quality and safety of a procedure. Although there has been progress in the administration of sedation by nurses in recent years, to our knowledge, the scientific evidence available in digestive endoscopy services has not been analyzed, which is a place where nursing professionals frequently administer sedation autonomously. Therefore, the objective of this study was to synthesize the scientific evidence available to date concerning the efficacy, safety and patient satisfaction of sedation administered by nurses during digestive endoscopies.

## 2. Materials and Methods

### 2.1. Design and Sources of Information

A systematic review was carried out following the guidelines of the PRISMA model (Preferred Reporting Items for Systematic Reviews and Meta-Analyses), which guarantees transparency and reproducibility in the selection, analysis and synthesis of the information reviewed [40]. This revision was registered in PROSPERO with the registration number CRD42025649246.

The databases consulted were PubMed, Scopus, SciELO, Web of Science and LILACS.

### 2.2. Search Strategies

The searches were carried out to directly answer the research question, which was developed in the population, intervention and outcome (PIO) formats (Table 1).

The research question was as follows: In patients undergoing digestive endoscopies (P), how does the administration of sedation by the nurse affect (I) in terms of efficacy, safety and patient satisfaction (O)?

The searches were carried out from November 2024 to March 2025. Scientific databases, which are presented in Table 2, were used to guarantee the completeness of the search.

### 2.3. Study Selection

The selection of the studies was carried out by two investigators, MHC and JMCT. For this purpose, the following inclusion and exclusion criteria were used:

Inclusion criteria:Studies involving patients undergoing digestive endoscopies with sedation administered by nurses.Articles published in the last five years.Research that evaluates the efficacy, safety, patient satisfaction or training of nurses in sedation

Exclusion criteria:Studies conducted on animals or simulations not applied to people.Opinion articles, editorials or nonsystematic reviews.Investigations without specific data on administered sedation.

To manage citations and eliminate duplicates, the Mendeley bibliographic manager (v1.19.8) was used. After eliminating duplicates, a preliminary selection was made by reading the title and abstract of the articles to determine those studies that would be read in depth. After these results were read in full, the studies that were part of the present systematic review were finally selected.

### 2.4. Quality Assessment

The quality of the selected articles was evaluated by three investigators (MHC, JMCT and JALA). Observational studies are more susceptible to bias than experimental studies are (particularly in terms of the allocation of participants and the evaluation of results). In this sense, the Joanna Briggs Institute (JBI) scale was used for observational studies. This scale consists of 9 items that evaluate different key methodological aspects that ensure the quality of the study. This is evaluated through a dichotomous scale of “yes” and “no”, and to allow greater flexibility when it is not possible to judge the fulfillment of an item definitively, or if it is not applied in the context of the evaluated study, the following are used: “unclear” or “not applicable” [41]. For case–control studies, the JBI tool for cases and controls was also used.

Following the recommendations of the Cochrane Collaboration to assess the risk of bias in clinical trials, the Risk of Bias 2 (RoB2) tool was used [42]. The RoB2 tool for RCTs consists of five domains that assess the risk of bias in (1) the randomization process, (2) planned interventions, (3) data loss, (4) measurement of variables of results and (5) selection of the reported results. Within these five domains, there are twenty-two different items whose answers can be one of the following six options: (a) yes, (b) most likely yes, (c) most likely not, (d) no, (e) not applicable (N/A) and (f) not informed (N/I). For each domain, the risk of bias was calculated. The total risk of bias was calculated as follows: (1) if the risk of bias was low in all domains, then the total risk was categorized as low; (2) if there was a risk of bias of some concern in any domain, the total risk was labeled “Some concern”; and (3) if there was a high risk of bias or multiple domains had some concern, the total risk of the study was categorized as high risk.

### 2.5. Data Extraction and Analysis

Data extraction was performed by two researchers: MHC and JMCT. From each selected study, the following data were collected: title and authors; year and country; study design; sample characteristics: sample size; objective of the study; main results: efficacy, safety (measured through complications) and patient satisfaction with sedation administered by nurses during digestive endoscopies; and conclusions.

All descriptive statistics were used, and a narrative synthesis was performed. To explore the complications of the use of sedation by procedures performed by nurses, the percentage of complications and the total number of events in the included studies were used. The data abstraction and synthesis processes were carried out independently by three reviewers. Disagreements were resolved by consensus. Owing to the methodology and heterogeneity of the included studies, it was not possible to perform a meta-analysis, so the results were presented as a narrative summary [43].

## 3. Results

### 3.1. Selection of Studies

According to the review, 292 articles were first identified through scientific databases such as PubMed (33), Scopus (66), SciELO (48), *Web of Science* (89) and LILACS (56). These articles were imported into the Mendeley reference manager to eliminate duplicates. A total of 145 unique articles remained, of which 30 were selected for more detailed screening, and 115 were excluded because they did not meet the initial criteria. Finally, 30 articles were considered eligible after an exhaustive analysis, 17 of which were excluded for the reasons explained in Figure 1. Therefore, 13 articles were included in this systematic review.

### 3.2. Characteristics of the Selected Studies

Among the total results, seven were retrospective observational studies [15,33,44,45,46,47,48], one was a retrospective cohort study [49], one was a case and control study [50], one was a randomized clinical trial (RCT) [34], and three prospective observational studies [19,27,51] were written in English. All patients who underwent digestive endoscopies, including colonoscopies, gastroscopies, echoendoscopies and esophagogastroduodenoscopies, were evaluated, and sedation was administered by the endoscopy nurse or anesthesia nurse. The most widely used drug is propofol. In total, the studies included a sample of 225,538 patients of legal age and 75 nurses, whose perceptions of the main complications arising during digestive endoscopies and whose training received in terms of sedation was analyzed, were included. Table 3 shows the relevant information on the selected studies.

### 3.3. Assessment of Risk of Bias

The RoB2 tool was used [42] to assess the risk of bias of the randomized clinical trials included in this systematic review [34], which revealed a moderate risk of bias. The risk of bias was low in most domains, but the lack of blinding in the intervention could have influenced the perception of the participants, especially in the measurement of satisfaction, a subjective result that may be biased by previous expectations. Figure 2 shows the risk of bias in the study items included in this review.

In the case of observational studies, the JBI scale was used for prevalence studies (52). Articles with fewer than five positive items (“yes”) were excluded from our review because of the high risk of bias. However, no studies were excluded for their methodological quality because they all showed a low to moderate risk of bias.

For the observational studies included in this review [15,19,27,33,44,45,46,47,48,51], the risk of bias is shown in Table 4. In general, the bias of the analyzed studies was moderate to low. These studies show variations in terms of the sampling frame and the size of the sample, but they show similarities in terms of the inclusion criteria as well as the statistical analyses used. All the studies described the inclusion criteria and demographics of the patients, and the response rate was generally good.

For the article cases and controls [50], the JBI scale was used [52] (Table 5). The study mentioned the comparability of the groups and the specific pairing of cases and controls in the study. Exposure and outcomes were measured in a standard way, suggesting a valid and reliable measurement in all groups, although it does not explicitly mention confounding factors or strategies to manage them. Thus, an appropriate statistical analysis was used.

Finally, for the retrospective cohort studies [49], the JBI scale was also used [52] (Table 6). This study has a low risk of bias.

### 3.4. Efficacy of Sedation Administered by Nurses in Digestive Endoscopies

Among the 13 articles included in the review, 9 address the efficacy of sedation administered by nurses in digestive endoscopies [27,33,34,44,45,46,48,49,50]. In accordance with these authors, sedation administered by nurses is effective in ensuring patient comfort and optimizing procedure times.

In accordance with Conigliaro et al. [33], the efficacy of the ED-BPS protocol (balanced sedation with propofol administered by non-anesthesiologists) reflects an adequate implementation of regulations and the continuous training of personnel, supported by regular audits. High patient tolerance and rapid psychomotor recovery not only increase patient satisfaction but also adherence to endoscopic procedures, which can increase early diagnosis rates in gastrointestinal diseases.

On the other hand, Lee et al. [45] underscore the efficacy of NAPCIS (Nurse-Administered Propofol Continuous Infusion Sedation) by offering adequate sedation in patients with complex conditions that make their management difficult, such as chronic substance use and post-traumatic stress disorder (PTSD). This approach allows sedative doses to be tailored according to individual needs, achieving success rates similar to those of patients without these conditions without compromising safety. Despite the higher requirements of fentanyl and propofol in some groups (for example, marijuana users), the results reinforce that NAPCIS not only optimizes the patient experience but also maintains low risks of adverse events, consolidating itself as an option practice in clinical settings. This method reduces the need for anesthesiologists in difficult-to-sedate patients, reducing costs and procedure time.

Furthermore, Steenholdt et al. [34] reported that deep sedation with propofol, administered by nurses, is an effective strategy to improve the patient experience during colonoscopy, especially in those with IBD (inflammatory bowel disease). NAPS patients reported significantly less pain and more amnesia, which positively influenced their willingness to undergo future procedures. These findings underscore that the choice of sedation protocol has a direct effect on the perception of the quality of medical care, which is especially relevant in chronic diseases such as IBD, where adherence to repeated colonoscopies is necessary.

### 3.5. Safety and Management of Complications During Sedation Administered by Nurses

Of all the studies selected, 12 evaluated the safety of sedation administered by nurses, as well as the management of complications during sedation [15,27,33,34,44,45,46,47,48,49,50,51]. In all the studies included in this review, nurses were authorized to independently administer sedative medications, including propofol, midazolam and fentanyl. Among these, propofol was the most frequently used medication across the majority of the studies. Additionally, in most of the studies included, nurses received specific training prior to administering propofol, frequently including ALS certification, although in some cases, the type of training was not specified.

Ten of the twelve studies reported low incidence rates of complications, which were minor complications, amounting to less than 2.5% [15,27,34,44,45,46,49,50,51]. Among the most frequent complications were transient desaturation, bradycardia and hypotension. All complications were easily resolved by the nurse. There were only moderate complications in two studies [33,51], with percentages of 0.001% and 0.05%, respectively.

According to Hidalgo-Cabanillas et al. [27], nurse-administered sedation with propofol is effective and safe for ensuring patient comfort and optimizing procedure times. The low incidence of serious adverse events, together with the successful management of minor complications, highlights the safety of this approach in patients classified as ASA I-II. This finding shows that nurses, by following a standardized protocol, can guarantee safe practices, even without the direct supervision of anesthesiologists. The authors also indicate the importance of establishing clear protocols and training staff in the use of sedatives, thus ensuring proper management of possible complications.

For Yamaguchi et al. [44], sedation during emergency endoscopy is safe and contributes to the proper management of patients with upper gastrointestinal bleeding. With the use of propofol, an effective and fast-acting sedative, patients’ agitation and body movements are significantly reduced, allowing endoscopists to perform procedures more easily and in less time. Furthermore, the absence of significant differences in the incidence of adverse events between the sedated and nonsedated groups supports the implementation of this approach in emergency situations, provided that the patient’s vital signs are closely monitored. According to these authors, these results reinforce the role of well-defined protocols and adequate training of nurses to administer sedation effectively, even in high-pressure settings such as emergency endoscopies.

According to Lee et al. [45], Nurse-Administered Propofol Continuous Infusion Sedation (NAPCIS) offers adequate and safe sedation in patients with chronic substance use and post-traumatic stress disorder (PTSD) without experiencing differences from patients without comorbidities of this type.

### 3.6. Patient Satisfaction and Experience with Nurse-Administered Sedation

Five studies analyzed the satisfaction and experience of patients who underwent digestive endoscopies with the administration of sedation by nurses [27,33,34,47,50]. User satisfaction with endoscopy nurse-administered sedation was high, indicating that patients are satisfied with the service received.

Monsma-Muñoz et al. [47] reported that a high level of patient satisfaction (4.27 out of 5) indicates that this care model is not only safe but also effective in guaranteeing a positive experience during colonoscopy. The positive perception of the procedure can be attributed to the adequate training of the nursing staff, as well as the application of a well-designed protocol, which minimized discomfort and optimized the clinical results.

Furthermore, Steenholdt et al. [22] reported that the level of satisfaction obtained with deep sedation could translate into greater compliance with endoscopic monitoring programs, thus optimizing long-term clinical results. These findings suggest that respecting patients’ preferences regarding the depth of sedation not only improves their experience but can also reduce anxiety and increase adherence to procedures, reinforcing the need to incorporate personalized sedation strategies into practice in clinics.

On the other hand, Hidalgo-Cabanillas et al. [27] reported that patient satisfaction is one of the fundamental pillars of care quality management programs; thus, it should be taken into account that satisfied patients better comply with the indications and follow-up and better tolerate the different treatments, whereas patients who are dissatisfied with the care received may suffer anxiety and stress, which makes them unable to respond correctly to the treatments that are proposed to them.

## 4. Discussion

This review analyzes the efficacy of the administration of sedation by digestive endoscopy nurses through different protocols established by different hospitals. Among the 13 studies included in this review, 9 address efficacy [27,33,34,44,45,46,48,49,50,51] and indicate that sedation administered by nurses is effective in ensuring patient comfort and optimizing procedure times [33,51]. Among the most frequent complications were transient desaturation, bradycardia and hypotension. These complications are usually the most frequent in the administration of sedation by nurses in digestive endoscopy services, as we have observed in different studies [29,47,53,54]. Interestingly, some studies have shown that nurses administer significantly lower sedative doses than anesthetists do, with fewer cardiopulmonary adverse events (2.2% vs. 4.7%) [49]. In fact, the rates of adverse events associated with the administration of sedation by anesthetists and/or intensivists in digestive endoscopies are similar to those reported in studies carried out by nurses and are even higher [55,56,57]. However, these studies were carried out before 2020 and may be justified given that, currently, the trend that is being followed in hospitals is the performance of this by nurses and endoscopists because the safety and efficacy rates are similar to those reported by anesthetists.

This low rate of incidence of adverse events, together with the successful management of minor complications, highlights the safety of the procedure; thus, sedation administered by nurses is a safe alternative to the exclusive supervision of anesthetists whenever adequate training and clear protocols exist. Therefore, the effectiveness of these procedures reflects an adequate implementation of regulations and the continuous training of personnel, supported by regular audits [33]. Our results revealed that the nurses used protocols such as the ED-BPS, NAPCIS and NAPS, all of which were effective.

Manno et al. (2021) confirmed that standardized protocols empower nurses to manage sedation safely, even without anesthesiologist oversight [51]. This highlights the importance of training in early complication recognition, especially for events such as hypoxemia or arrhythmias.

On the other hand, a recent study in which nurses performed sedation in endoscopy services highlighted the relationship between the quality of training for nurses and safety in endoscopic procedures [19]. Although most nurses had more than two years of experience in the service, the lack of specific prior training in sedation points to a critical deficiency that may influence the proper management of respiratory and cardiovascular emergencies during procedures. Therefore, it is important to determine systematic and practical educational programs that include realistic simulations to prepare nursing personnel in the management of emergencies related to sedation [19].

Higher education levels among nursing staff correlate with improved outcomes and cost-effectiveness, including a 10.7% reduction in hospital mortality [58]. These findings align with broader healthcare trends emphasizing advanced nursing competencies as a strategy for resource optimization and quality improvement [59]. Multiple studies in this review echo the importance of protocolized and continuous education [19,27,47,49,51,60].

On the other hand, the high level of patient satisfaction shows that this care model is not only safe but also effective in guaranteeing a positive experience during colonoscopy [29,47,61]. The positive perception of the procedure can be attributed to the adequate training of the nursing staff, as well as the application of a well-designed protocol, which minimized discomfort and optimized the clinical results. This care model could be extended to more complex procedures in the future, as long as an approach focused on continuous training and the integration of multidisciplinary teams is maintained [47]. In addition, the level of satisfaction obtained with deep sedation could translate into greater compliance with endoscopic monitoring programs, thus optimizing long-term clinical results [34].

Importantly, one of the studies included in this review, conducted by Yamaguchi et al. [44], was conducted in an emergency setting. Emergency endoscopy significantly differs from elective procedures, as it often involves critically ill patients with multiple comorbidities, increasing the risk associated with sedation. In such contexts, the presence and supervision of an anesthesiologist may be necessary to ensure patient safety [18,61]. Therefore, this case should be interpreted separately from the rest of the studies included in the review.

According to Andrade et al. (2021) [62], the use of propofol for sedation in critically ill patients is not only safer but also more cost-effective than the use of midazolam. This translates into an improvement in the quality of care and a reduction in the economic burden for health systems. For this reason, the relationship between cost-effectiveness and the administration of sedation by nurses plays an important role in the implementation of efficient care models. The possibility of incorporating protocols that allow nurses to safely administer propofol could further reduce the costs associated with prolonged ICU stays and related adverse events. Other studies [1] have discussed other alternative drugs to propofol, such as fospropofol and ciprofol, which have similar characteristics in terms of cost-effectiveness.

In addition, the administration of sedation by nurses has proven to be cost-effective, especially in high-volume patient settings. The costs associated with sedation also include personnel, medications, monitoring equipment and the time required for induction and recovery. Consequently, medical centers seek to maximize the efficiency of available resources without compromising patient safety [15,49]. The use of certain sedatives, such as propofol administered in adequate doses by trained nurses, can shorten recovery times and improve efficiency in endoscopy units, benefiting both the health system and the patient [49,50].

### Limitations and Strengths

Regarding the limitations, it should be noted that only one RCT in the literature could be included, so a meta-analysis could not be carried out, as observational studies of a control group were lacking. Most studies on this topic in the literature are prospective and retrospective in nature, with the latter being more frequent; thus, more RCTs should be conducted on this topic.

In terms of strengths, this systematic review is current because it presents the knowledge and results of studies published in the last 5 years. Notably, although there are different forms of sedation, the results in terms of efficacy, safety and satisfaction have always followed the same procedure. Another strength of this study is the representative sample of patients who were sedated by a nurse when they underwent a digestive endoscopy.

Finally, a similar procedure was used in all the intervention studies, and there were only slight modifications in the doses and methods of drug administration used. To our knowledge, there are no updated systematic reviews evaluating the efficacy, safety and satisfaction of patients undergoing gastrointestinal endoscopy with sedation by a nurse.

## 5. Conclusions

The results of this study suggest that nurse-administered sedation demonstrates high efficacy. The evidence confirms that nurse-administered propofol sedation is safe in endoscopic procedures, with low rates of adverse events.

According to our results, the patients reported high levels of satisfaction, little discomfort and improved perceptions of the quality of the procedure with nurse sedation. Additionally, formal training and certification are fundamental pillars to ensure safety in the administration of sedation by nurses.

Finally, nurse-administered sedation is a cost-effective alternative.

## Figures and Tables

**Figure 1 diagnostics-15-01030-f001:**
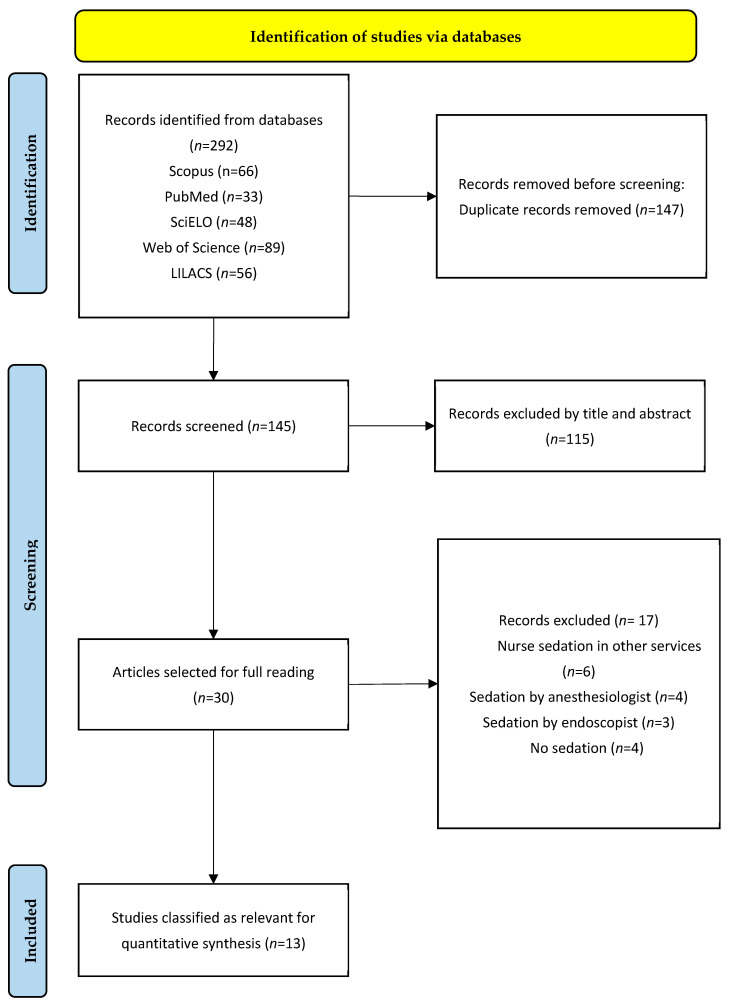
PRISMA flow diagram.

**Figure 2 diagnostics-15-01030-f002:**
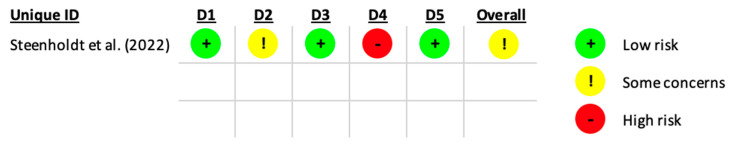
Risk of bias for RCTs [34].

**Table 1 diagnostics-15-01030-t001:** PIO question.

Population	Intervention	Results
Patients undergoing digestive endoscopies	Nurse-administered sedation	Efficacy, safety and patient satisfaction in nurse-administered sedation

**Table 2 diagnostics-15-01030-t002:** Search strategy.

Database	Keywords	Search String
PubMed	Sedation, endoscopy, nurses	(Sedation AND endoscopy AND nurses) OR (Complications AND sedation AND endoscopy)
Scopus	Safety, complications, management	(“Sedation administered by nurses” AND “complications” OR “management”)
SciELO	Training, cost-effectiveness, satisfaction	((Nurse-administered sedation) AND (patient satisfaction OR safety)) AND NOT pediatrics
*Web of Science*	Evaluation, rates, protocols	(Nurse AND sedation AND “endoscopic procedures” NOT “pediatric cases”)
LILACS	Nurses, endoscopy, quality	(MeSH terms: “Nurse-administered sedation” AND “Endoscopy”)

**Table 3 diagnostics-15-01030-t003:** Results table.

Author/ Year	Design	Participants	Nurses Received ALS Training	Country	Results	Conclusions
Conigliaro et al. (2024) [33]	Retrospective observational	N = 14,415	Yes	Italy	The results showed low rates of adverse events (6% hypotension, 0.9% bradycardia) and a minimal percentage (0.01%) of serious complications that required intubation.	Nurse-administered sedation directed by endoscopists is safe and effective in low-risk patients, standing out as a viable option in routine endoscopies.
Lee et al. (2021) [45]	Retrospective observational	N = 1427	Not specified	USA	The results indicated that patients with difficulties being sedated required higher doses of fentanyl and propofol, but the success rates of the procedures were high (95.1–100%), with similar procedure and recovery times compared to the control group. Adverse events were rare (26 cases), and no serious complications or deaths were recorded.	NAPCIS proved to be a safe and effective alternative to provide sedation in patients considered difficult to sedate.
Yamaguchi et al. (2023) [44]	Observational Retrospective divided into two groups with and without sedation	N = 171	Not specified	Japan	The results show that the sedation group had a significantly shorter procedure time (17.6 vs. 20.2 min, *p* = 0.04), with similar success rates in both groups. The complication rate was hypotension 3%, bradycardia 2.3% and hypoxia 1.5%. No significant differences were found in the incidence of adverse events or mortality.	The sedation administered during emergency endoscopies by HDA is safe; in addition, it reduces the time of the procedure and does not increase the risk of complications.
Sato et al. (2019) [46]	Observational Retrospective	N = 150.211	Not specified	Japan	It was observed that the mean dose of propofol used was 77 mg for EGD and 99 mg for colonoscopy. As an adverse event, only 1.3% of patients required supplemental oxygen temporarily (1.4% for EGD and 0.8% in colonoscopy), with no cases of mechanical ventilation, severe hypotension or mortality.	Nurse-administered mono-sedation with propofol, using doses of less than 200 mg, has been shown to be a safe and effective approach for outpatient gastrointestinal endoscopies.
Monsma-Muñoz et al. (2022) [47]	Retrospective observational	N = 381	Not specified	Spain	A total of 5% of the patients experienced oxygen desaturation (<90%), without requiring mask ventilation, while 7.35% presented hypotension and 3.94% bradycardia. In 22% of the cases, consultation with the supervising anesthesiologist was necessary. Patient satisfaction, evaluated on a scale of 1 to 5, reached an average of 4.27, and perceived pain was minimal according to a numerical verbal scale.	Nurse-administered sedation following a consensual protocol proved to be safe and effective in low-risk patients, reinforcing the viability of this care model.
Steenholdt et al. (2022) [34]	ECA	N = 130	Not specified	Denmark	Patients with deep sedation had a significantly higher mean score (60.1 vs. 51.2; *p* < 0.001) due to less pain, more amnesia, greater pleasure with sedation and a positive experience compared to previous sedations. In addition, this group showed a greater willingness to undergo future colonoscopies with the same protocol. In terms of safety, no patient with NAPS presented desaturation <92%, in contrast to six cases in the moderate sedation group.	Deep sedation with propofol significantly improves patient satisfaction and may increase adherence to endoscopic monitoring programs in patients with IBD.
Hidalgo-Cabanillas et al. (2024) [19]	Prospective observational	N = 75	Yes 64.2%	Spain	The results showed that 100% of the nurses were currently using sedation, but only a minority had received specialized training in sedation or advanced life support. In addition, significant differences were found in the availability of resources between hospitals.	The training of nurses who perform sedation is insufficient, and the variability in resources and standards between hospitals suggests the need to implement continuous training programs, regulated at the institutional level, to improve the safety and effectiveness of the procedures.
Manno et al. (2021) [51]	Prospective observational	N = 10,755	Yes	Italy	All staff (doctors and nurses) completed the ESGE-ESGENA sedation course. In total, 12,132 patients underwent endoscopic procedures, 10,755 (88.6%) of which were performed in a nonanesthesiological setting. Of these, approximately 20% used moderate sedation with midazolam + fentanyl, and 80% used deep sedation with additional propofol. There were no sentinel adverse events, 5 (0.05%) of moderate risk and 18 (0.17%) of lower risk, all during moderate or deep sedation, and all managed by endoscopy personnel without the need for assistance from the anesthesiologist.	After completing the ESGE-ESGENA sedation training program, the rate of adverse events was very low. The findings support the implementation of the program in all digestive endoscopy units and its inclusion in the curriculum for physicians and nurses to ensure safe endoscopic procedures.
Hidalgo-Cabanillas et al. (2024) [27]	Prospective observational	N = 660	Not specified	Spain	Most of the patients indicated great satisfaction, especially valuing the care provided by the nurses. However, they negatively highlighted the waiting time for the appointment and the wait on the same day. The incidence of complications was minimal, with only 2% of cases, most of them transient desaturations.	The sedation administered by nurses in digestive endoscopy is effective and safe; this is because it shows high satisfaction and low complication.
McKenzie et al. (2021) [15]	Observational Retrospective	N = 18,910	Yes	USA	The final sample size was 18,910 colonoscopy procedures and EGD procedures. In both colonoscopy and EGD procedures, there were no major adverse events. Mild adverse event rates were low (hypoxia 2.2%, hypotension 2.6% (32p) and bradycardia 4.4%) in both types of procedure and were not different between patients with ASA I/II and ASA III.	The EDNAPS is safe in both ASA I/II patients and ASA class III patients undergoing routine outpatient endoscopy.
Gururatsakul et al. (2021) [48]	Observational Retrospective	N = 24,958	Yes	Australia	During the 78-month period, a total of 24,958 procedures were analyzed with EDNAPS. Of these, 9539 were ASA 1 (38.2%), 13,680 were ASA 2 (54.8%), 1733 were ASA 3 (6.9%) and 4 were ASA 4 (0.02%). Complications related to sedation occurred in 66 patients (0.26%), predominantly transient hypoxic episodes. No patient required intubation of the airway, and there was no mortality related to sedation. Complications related to sedation increased with the ASA class and were significantly more frequent with gastroscopy.	Propofol sedation administered by endoscopic nurses is a safe way to perform endoscopic sedation in low-risk patients in the hospital setting.
Tiankanon et al. (2020) [49]	Retrospective cohort	N = 189	Yes	Thailand	A total of 278 eligible patients were included. There were 189 patients in the NAPS group and 63 in the OAPS group for analysis. Demographics were not different between the two groups. All procedures were technically successful with no difference in cecal intubation time. The dose of propofol/ kg/hour was significantly lower in the NAPS group (11.4 ± 4 mg/kg/hour versus 16.6 ± 8 mg/kg/hour; *p* < 0.001). There were fewer minor cardiopulmonary adverse events in NAPS compared to the OAPS group (2.2% vs. 4.7%; *p* = 0.014).	NAPS in elective colonoscopy in low-risk patients is as effective as OAPS but requires a significantly lower dose of propofol. Minor cardiopulmonary adverse events were recorded in the NAPS group compared to OAPS.
Lin et al. (2021) [50]	Cases and controls	N = 3331	Yes	USA	The success rates of the NAPCIS procedures were high (99.1–99.2%) and similar to those observed in the CAPS (98.8–99.0%) and MF (99.0–99.3%) controls. The recovery times of NAPCIS were shorter than those of CAPS and MF. Validated physician and patient satisfaction scores were generally higher for NAPCIS subjects compared to CAPS and MF subjects. For NAPCIS, there were only four cases of oxygen desaturation and no serious complications related to sedation. These low complication rates were similar to those observed with CAPS (eight cases) and MF (three cases).	NAPCIS appears to be a safe, effective and efficient means of providing moderate sedation for upper endoscopy and colonoscopy in low-risk patients.

**Table 4 diagnostics-15-01030-t004:** JBI for prevalence studies.

		Conigliaro et al. (2024) [33]	Lee et al. (2021) [45]	Yamaguchi et al. (2023) [44]	Sato et al. (2019) [46]	Monsma-Muñoz et al. (2022) [47]	Hidalgo-Cabanillas et al. (2024) [19]	Manno et al. (2021) [51]	Hidalgo Cabanillas et al. (2024) [27]	McKenzie et al. (2021) [15]	Gururatsakul et al. (2021) [48]
**Was the sample frame appropriate to address the target population?**											
**Were study participants sampled in an appropriate way?**											
**Was the sample size adequate?**											
**Were the study subjects and the setting described in detail?**											
**Was the data analysis conducted with sufficient coverage of the identified sample?**											
**Were valid methods used for the identification of the condition?**											
**Was the condition measured in a standard, reliable way for all participants?**											
**Was there appropriate statistical analysis?**											
**Was the response rate adequate, and if not, was the low response rate managed appropriately?**											
		Yes									
		No									
		Unclear									
		Not applicable									

**Table 5 diagnostics-15-01030-t005:** JBI for case–control studies.

		Lin et al. (2021) [50]
**Are the groups comparable other than the presence of disease in cases or the absence of disease in controls?**		
**Are cases and controls matched appropriately?**		
**Are the same criteria used for identification of cases and controls?**		
**Was exposure measured in a standard, valid and reliable way?**		
**Was exposure measured in the same way for cases and controls?**		
**Were confounding factors identified?**		
**Were strategies to deal with confounding factors stated?**		
**Were outcomes assessed in a standard, valid and reliable way for cases and controls?**		
**Was the exposure period of interest long enough to be meaningful?**		
**Was appropriate statistical analysis used?**		
	Yes	
	No	
	Unclear	
	Not applicable	

**Table 6 diagnostics-15-01030-t006:** JBI for prevalence studies.

		Tiankanon et al. (2020) [49]
**Was the sample frame appropriate to address the target population?**		
**Were study participants sampled in an appropriate way?**		
**Was the sample size adequate?**		
**Were the study subjects and the setting described in detail?**		
**Was the data analysis conducted with sufficient coverage of the identified sample?**		
**Were valid methods used for the identification of the condition?**		
**Was the condition measured in a standard, reliable way for all participants?**		
**Was there appropriate statistical analysis?**		
**Was the response rate adequate, and if not, was the low response rate managed appropriately?**		
	Yes	
	No	
	Unclear	
	Not applicable	

## Data Availability

Data sharing is not applicable. No new data were created or analyzed in this study.

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
