# Peer review of "Nurse-Administered Sedation in Digestive Endoscopy: A Systematic Review"

_diagnostics, 2025, doi:10.3390/diagnostics15081030_

Round 1

Reviewer 1 Report

Comments and Suggestions for Authors

In the present systematic review Hidalgo et al summarized current knowledge about sedation administered by nurses during digestive endoscopy. Main comments:

1) Please check “y” in the title

2) Why was the timeframe of search limited to Nov 24 to Mar 25? It is too much narrow to enclose all relevant literature, it should be widened.

3) Lines 150-154: PIO or IOP?

4) Authors should clearly state whether all nurses were allowed to administer propofol, midazolam or fentanyl.

5) Did all nurses receive an ALS training before giving propofol?

6) Please check figure 1, as only one study has been reported. I do not understand the meaning of such image, as risk of bias was reported in tables 4-6.

7) As I understand by reading table 3, one study (Yamaguchi) was performed in emergency setting. This deserves a separate discussion, as emergency endoscopy is completely different, with patients with relevant comorbidities, therefore anaesthesiologists care should be always guaranteed.

Comments on the Quality of English Language

See point 1 (title)

Author Response

RESPOND TO REVIEWER

Ref:  diagnostics-3554201. “Nurse-Administered sedation in digestive endoscopy: a systematic review”.

We appreciate very much your constructive comments, helpful information and your time. We have considered all suggestions and incorporated them into the revised manuscript, and as a result, we believe our manuscript is stronger. Responses to his comments are written in bold type. We have highlighted in yellow the changes made to the manuscript.

REVIEWER 1:

1) Please check “y” in the title

Thank you for your comments. We have modified the title.

2) Why was the timeframe of search limited to Nov 24 to Mar 25? It is too much narrow to enclose all relevant literature, it should be widened.

We appreciate your observation. The mention of the timeframe "from November 2024 to March 2025" refers to the period during which the authors conducted the literature search, not to the publication dates of the articles included. The actual search was limited to studies published in the last five years (2019–2024) to ensure the inclusion of the most recent and relevant evidence. We acknowledge that the original phrasing may have caused confusion and will clarify this point in the revised manuscript.

3) Lines 150-154: PIO or IOP?

Thank you for your comment. We have modified this information, is PIO question.

4) Authors should clearly state whether all nurses were allowed to administer propofol, midazolam or fentanyl.

Thank you for your comment. In all the studies included in this review, nurses were authorized to independently administer sedative medications, including propofol, midazolam, and fentanyl. Among these, propofol was the most frequently used medication across the majority of the studies. We have added the information in the result section, in “3.5 safety and management of complications during sedation administered by nurses”.

5) Did all nurses receive an ALS training before giving propofol?

Thank you for your comment.  Most of the studies included in this review reported that nurses received specific training before administering propofol, and many of them explicitly mentioned Advanced Life Support (ALS) certification. However, some studies did not detail the exact type of training provided. This point has now been clarified in the revised manuscript to reflect the variability in the information reported.

6) Please check figure 1, as only one study has been reported. I do not understand the meaning of such image, as risk of bias was reported in tables 4-6.

Thank you for your comment. Figure 1 represents the risk of bias assessment specifically for the study by Steenholdt et al., which is a randomized clinical trial. For this study, we applied the RoB 2 tool, as it is the recommended instrument for evaluating bias in randomized trials. In contrast, Tables 4 to 6 present the risk of bias assessments for the remaining included studies, which are observational in nature (prevalence and case-control designs). For these, we used the appropriate JBI critical appraisal tools. We have revised the manuscript to clarify this distinction.

7) As I understand by reading table 3, one study (Yamaguchi) was performed in emergency setting. This deserves a separate discussion, as emergency endoscopy is completely different, with patients with relevant comorbidities, therefore anaesthesiologists care should be always guaranteed.

Thank you for your comment.   You are correct in pointing out that the study by Yamaguchi et al. was conducted in an emergency setting, which indeed differs significantly from elective procedures. Emergency endoscopy often involves patients with acute conditions and multiple comorbidities, where the clinical complexity is greater and the risks associated with sedation are higher. For this reason, we agree that the presence of an anaesthesiologist may be necessary in such settings to ensure patient safety.

We have included a specific comment in the discussion section of the manuscript to highlight this distinction and to acknowledge the unique considerations related to sedation in emergency endoscopy.

Reviewer 2 Report

Comments and Suggestions for Authors

Dear authors,

Thank you for the opportunity to review this interesting review.

The topic is timely and clinically relevant, especially as the role of nurses continues to evolve in procedural sedation practices. Overall, the manuscript demonstrates a commendable effort to synthesize the current evidence on efficacy, safety, and patient satisfaction associated with nurse-administered sedation. However, there are several areas that would benefit from revision to strengthen clarity, scientific rigor, and overall presentation.

  • The discussion is too descriptive and lengthy in places. Please reduce repetition and synthesize the results more concisely. Compare your findings more critically with existing literature beyond the included studies.
  • Remove unnecessary spaces between paragraphs and write the authors' names and surnames in non-capitalized form
  • The abstract is too long and contains methodological details that may be better placed in the Methods section. Please streamline.
  • The conclusions are too long. Summarize your conclusions to provide a take home message in 5 lines or less.
Comments on the Quality of English Language

The English language is generally understandable but would benefit from professional editing to improve grammar, sentence structure, and flow. Some awkward phrasing appears throughout (e.g., “the nurse and the patient… health. well-being”).

Author Response

RESPOND TO REVIEWER

Ref:  diagnostics-3554201. “Nurse-Administered sedation in digestive endoscopy: a systematic review”.

We appreciate very much your constructive comments, helpful information and your time. We have considered all suggestions and incorporated them into the revised manuscript, and as a result, we believe our manuscript is stronger. Responses to his comments are written in bold type. We have highlighted in yellow the changes made to the manuscript.

REVIEWER 2:

  • Dear authors,

Thank you for the opportunity to review this interesting review.

The topic is timely and clinically relevant, especially as the role of nurses continues to evolve in procedural sedation practices. Overall, the manuscript demonstrates a commendable effort to synthesize the current evidence on efficacy, safety, and patient satisfaction associated with nurse-administered sedation. However, there are several areas that would benefit from revision to strengthen clarity, scientific rigor, and overall presentation.

Thank you for your positive comments.

  1. The discussion is too descriptive and lengthy in places. Please reduce repetition and synthesize the results more concisely. Compare your findings more critically with existing literature beyond the included studies.

Thank you for your comment. We have revised and restructured the discussion section to make it more concise and avoid unnecessary repetition. We believe these changes strengthen the discussion and improve the quality of the manuscript.

  1. Remove unnecessary spaces between paragraphs and write the authors' names and surnames in non-capitalized form

Thank you for your comment. We have eliminated unnecessary spaces between paragraphs and have written the authors' names and surnames in non-capitalised format.

  1. The abstract is too long and contains methodological details that may be better placed in the Methods section. Please streamline.

Thank you for your comment. We have simplified the abstract.

  1. The conclusions are too long. Summarize your conclusions to provide a take home message in 5 lines or less.

Thank you for your comment. We have summarized the conclusions.

  1. The English language is generally understandable but would benefit from professional editing to improve grammar, sentence structure, and flow. Some awkward phrasing appears throughout (e.g., “the nurse and the patient… health. well-being”).

Thank you for your comment. We have improved the English of the article to make it more understandable. To ensure clarity and linguistic quality, the manuscript has been edited by a professional editing service.

Round 2

Reviewer 1 Report

Comments and Suggestions for Authors

Regarding point 5, it is important to include in table 3 in which studies nurses receivel ALS training.

All other answers were fine.

Author Response

RESPOND TO REVIEWER

Ref:  diagnostics-3554201. “Nurse-Administered sedation in digestive endoscopy: a systematic review”.

We appreciate very much your constructive comments, helpful information and your time. We have considered all suggestions and incorporated them into the revised manuscript, and as a result, we believe our manuscript is stronger. Responses to his comments are written in bold type. We have highlighted in yellow the changes made to the manuscript.

REVIEWER 1:

Regarding point 5, it is important to include in table 3 in which studies nurses receivel ALS training.

All other answers were fine.

Thank you for your comments. We have added a new section in table 3: “Nurses received ALS training” indicating whether nurses received ALS training.

Reviewer 2 Report

Comments and Suggestions for Authors

Thank you to replied to my comment.

I think the conclusion is too long.

Author Response

RESPOND TO REVIEWER

Ref:  diagnostics-3554201. “Nurse-Administered sedation in digestive endoscopy: a systematic review”.

We appreciate very much your constructive comments, helpful information and your time. We have considered all suggestions and incorporated them into the revised manuscript, and as a result, we believe our manuscript is stronger. Responses to his comments are written in bold type. We have highlighted in yellow the changes made to the manuscript.

REVIEWER 2:

Thank you to replied to my comment.

I think the conclusion is too long.

Thank you for your comment regarding the length of the conclusion. We have carefully reviewed the text and summarized the conclusion. We believe that the current length is necessary to adequately summarize the main findings, answer the research question and highlight the practical implications of the study.